# Focus & Gating: A Multimodal Approach for Unveiling Relations in Noisy Social Media

## ABSTRACT

With the rise of multimedia-driven content on the internet, multimodal relation extraction has gained significant importance in various domains, such as intelligent search and multimodal knowledge graph construction. Social media, as a rich source of image-text data, plays a crucial role in populating knowledge bases. However, the noisy information present in social media data poses a challenge in multimodal relation extraction. Current methods focus on extracting relevant information from images to improve model performance but often overlook the importance of global image information. In this paper, we propose a novel multimodal relation extraction method, named FocalMRE, which leverages image focal augmentation, focal attention, and gating mechanisms. FocalMRE enables the model to concentrate on the image's focal regions while effectively utilizing the global information in the image. Through gating mechanisms, FocalMRE optimizes the multimodal fusion strategy, allowing the model to select the most relevant augmented regions for overcoming noise interference in relation extraction. The experimental results on the public MNRE dataset reveal that our proposed method exhibits robust and significant performance advantages in the multimodal relation extraction task, especially in scenarios with high noise, long-tail distributions, and limited resources.

## CCS CONCEPTS

• **Computing methodologies** → **Information extraction**; • **Information systems** → **Information extraction**.

## KEYWORDS

multimodal relation extraction, focal augmentation, focal attention, gating mechanism, noisy social media

## 1 INTRODUCTION

With the rapid advancement of multimedia technologies, internet data has shifted from text-centric to multimedia-driven content. As such, multimodal relation extraction tasks have become increasingly significant across various domains, including intelligent search [16, 32], question answering [13, 24, 27], personal recommendation [21, 22], and knowledge graph construction [17, 25]. Social media, in particular, is a vital information source that harbors a plethora of image-text data containing rich information crucial

**Unpublished working draft. Not for distribution.**

for populating knowledge bases. Zheng et al. [36] introduced the first multimodal relation extraction dataset collected from social media (as illustrated in Figure 1-a), and numerous studies have built upon this foundation since then, providing valuable insights and prompting the development of this research task.

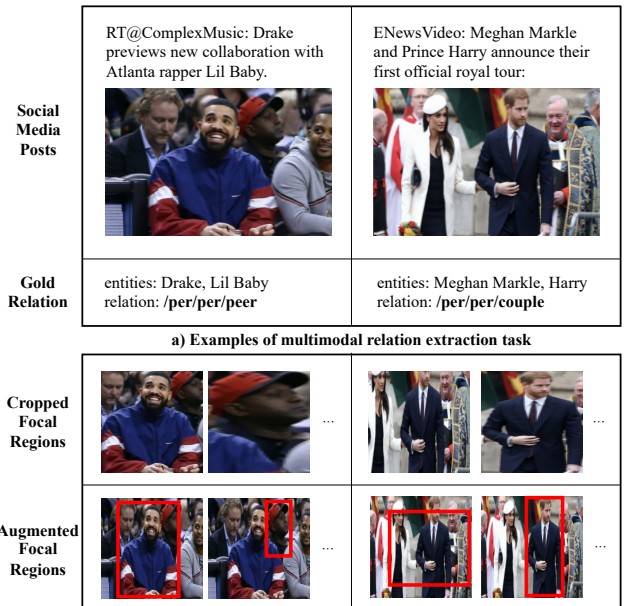

a) Examples of multimodal relation extraction task

b) Different processing methods for the focal region of the original image: cropping or augmentation

**Figure 1: Two cases illustrating multimodal relation extraction and focal region processing methods.**

However, a salient feature of social media data is the prevalence of noisy information, such as irrelevant backgrounds and objects, which poses a significant challenge in multimodal relation extraction. Current methods primarily aim to extract information relevant to the text from images to improve model performance. For instance, focal regions in images are cropped using object detection and visual grounding techniques and resized to a uniform size for model input. Nonetheless, these approaches overlook the importance of global image information crucial for understanding the relative positions, spatial relationships, and overall context of objects in the image, which can enhance the accuracy of relation extraction. As depicted in Figure 1-b, the cropped regions that are inputted into the model ultimately lose the actual size and relative position relationships of the objects in the source image. This loss of information renders it impossible to accurately determine the primary and secondary relationships between the objects. Therefore, preserving global image information while focusing on the image regions most relevant to the text is crucial.

To address this challenge, this paper proposes a multimodal relation extraction method based on image focal augmentation, focal attention, and gating mechanisms. The method leverages global image information and enables the model to focus on image regions closely related to the text for more accurate relation extraction. Specifically, the original image is augmented with focal regions (as illustrated in Figure 1-b), and the attention from text to the image with focal regions is calculated to obtain a multimodal representation. On this basis, the gating mechanism filters irrelevant image content to noise, allowing the model to adaptively select information useful for relation extraction, reducing the impact of errors from focal region detection models.

In summary, our contribution can be summarized as three-fold:

- Firstly, we propose FocalMRE, a novel multimodal relation extraction method designed to address the challenges of noisy data in social media. FocalMRE utilizes image focal augmentation, focal attention, and gating mechanisms to concentrate on the image's focal regions while incorporating global image information, which is often overlooked by existing methods.
- Secondly, we employ focal augmentation and gating mechanisms to optimize the multimodal fusion strategy. This allows the model to selectively emphasize the most relevant augmented regions, thereby mitigating noise interference and improving the accuracy of relation extraction.
- Finally, we validate the effectiveness of FocalMRE on the public MNRE dataset through comprehensive experiments, demonstrating that FocalMRE achieves robust and significant performance improvements, particularly in high-noise, long-tail, and low-resource scenarios. This highlights the method's effectiveness in multimodal relation extraction task, showcasing its potential in practical applications.

## 2 RELATED WORKS

In the realm of artificial intelligence, there is an increasing focus on multimodal relation extraction research. The objective of this task is to identify relationships between entities using multimodal data comprising both text and images. Zheng et al. [36] were the pioneers in proposing this task and highlighted that conventional text-based relation extraction models underperform in scenarios, such as social media, where text content is limited and short. To tackle this challenge, they developed the first social media-based multimodal relation extraction dataset, MNRE, and proposed multiple multimodal baselines. By leveraging multimodal information, these baselines enhance the accuracy of relation extraction while providing valuable data support for future research.

Subsequent research has primarily concentrated on effectively integrating textual and visual information. MEGA [35] introduced graph structural alignment and semantic alignment, achieving precise alignment of entity relations in text and images by comparing the structural and semantic similarities between visual scene graphs and textual syntactic dependency graphs. Li et al. [14] conducted experiments by shuffling the image-text pairs in the dataset and demonstrated that fine-grained alignment mechanisms effectively use visual information to enhance the accuracy of relation extraction. Xu et al. [28] used reinforcement learning to reorganize

segmented datasets intelligently and discovered that different data types are suitable for different processing models, with some being more appropriate for multimodal models and others for pure textual models. These studies reveal the impact of fine-grained structural and semantic alignment mechanisms.

Moreover, researchers have explored the incorporation of external knowledge. MoRe [26] designed text and image-based retrieval modules to retrieve external knowledge and integrated them into textual and visual task models for predictions. For the final decision, predictions are combined using a Mixture of Experts (MoE) module, thereby augmenting the model's input information and improving the accuracy of entity relation prediction. Hu et al. [12] proposed a novel pre-training method using unlabeled image-caption pairs that aligns entity-object and relation-image. They generated soft pseudo-labels for these alignments, employed them as self-supervised signals for pre-training, thereby enhancing the model's ability to extract entities and relations. TMR [34] employed generative back-translation using diffusion models to create pseudo-parallel data and trained a high-resource estimator to generate fine-grained alignment scores, effectively modeling the misalignment between text and images, outperforming previous state-of-the-art methods. He et al. [11] explored ways of encoding textual attributes, visual depth, and object positions that could tackle the problems of semantic inconsistencies and multi-object ambiguity, and finally enhance the performance of multimodal relation extraction.

Lastly, researchers have addressed the impact of image modality noise. UMGF [33] represented input sentences and images as a unified multi-modal graph, capturing semantic relationships between multi-modal semantic units. MKGformer [6] combined object detection and visual grounding models to extract image regions related to text, reducing noise interference from irrelevant images effectively. They proposed a multi-level fusion method that tightly integrates visual and textual representations through coarse-grained and fine-grained interaction and fusion, significantly enhancing performance.

With the rapid advancement of large language models, there has been a rise in multimodal relation extraction methods that depend on these models. Chen et al. [5] introduced a method that employs chain-of-thought prompt distillation. This technique utilizes chain-of-thought to demonstrate the reasoning process explicitly and facilitates the transfer of knowledge from large-scale pre-trained models to target task models through distillation. Additionally, Cai et al. [4] put forth a novel few-shot multimodal named entity recognition (FewMNER) task. They have accomplished accurate multimodal named entity recognition in few-shot conditions by creating effective demonstrations that merge task instructions and entity category definitions.

## 3 METHODOLOGY

### 3.1 Problem Formulation

In multimodal relation extraction, the objective is to predict relations between entities based on both textual and visual inputs. This task can be modeled using a function $F = (e_1, e_2, S, V) \rightarrow \mathcal{R}$, where $e_1$ and $e_2$ represent the pre-extracted textual entities. Given a sentence $S$ with marked entities $e_1$ and $e_2$ and the visual content

$V$, the goal is to classify the corresponding relation tag $\mathcal{R}$ between $e_1$ and $e_2$.

## 3.2 Overall Architecture

In this paper, we introduce a novel multimodal relation extraction model named **FocalMRE**. The framework of this model is shown in Figure 2. It is based on the results of object detection and visual grounding to enhance the focus of images by highlighting focal regions. The focal augmented images and text content are then encoded separately by unimodal encoders to form their respective representations. To achieve a deep integration of visual and textual features, we introduce a multimodal fusion layer that incorporates focal attention and gating mechanisms, resulting in a multimodal representation. Finally, the model extracts the entity features required and accurately predicts the relations through a classifier.

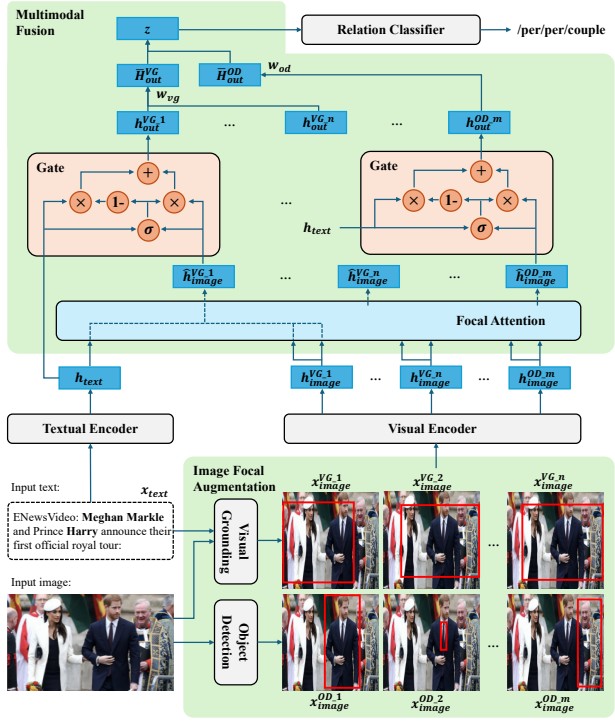

**Figure 2: The framework of FocalMRE.**

**Textual Encoder.** The textual encoder in our model is based on the BERT model [8]. It is responsible for extracting deep textual representations from the input token sequence, which consists of text units such as words, phrases, or symbols. First, an embedding layer transforms the token sequence into embedding vectors in a high-dimensional vector space, with positional embeddings added to capture the positional information within the sequence. The text encoder adopts a multi-layer structure, with each layer composed of a multi-head self-attention mechanism (MHA) and a feedforward neural network (FFN). MHA allows the model to dynamically attend to the dependencies between different tokens, capturing contextual

information in the text. FFN performs non-linear transformations on the MHA outputs to further extract textual features. The use of residual connections and layer normalization techniques effectively mitigates the vanishing gradient problem in deep neural networks and stabilizes the model training process. Through layer-by-layer computation, the text encoder outputs the hidden states of each token, which contain rich contextual information and features of the text, laying a solid foundation for subsequent multimodal fusion.

$$h_{text} = \text{TextualEncoder}(x_{text}) \tag{1}$$

**Visual Encoder.** The visual encoder adopts the Vision Transformer (ViT) model [9]. The process begins by resizing each focal augmented image to ensure input consistency. The image is then split into several patches, each of which is embedded into a high-dimensional space through linear projection to form patch embeddings. Positional embeddings are added to distinguish information from different locations. These patch embeddings are then sent to ViT for further processing. In ViT, the visual embeddings are encoded through a multi-head self-attention mechanism and feedforward neural networks, with multiple Transformer layers stacked to progressively update the initial embeddings, thereby extracting high-level feature representations of the image. The output of the visual encoder is the feature representation of the focal augmented image, which not only reinforces visual information from the focal regions but also retains global information, such as spatial relationships within the image, providing strong support for subsequent multimodal fusion.

$$h_{image} = \text{VisualEncoder}(x_{image}) \tag{2}$$

## 3.3 Image Focal Augmentation

In this paper, we propose a simple method to augment image focus by constructing focal frames. This approach directly augments the original image, reducing reliance on external knowledge, enhancing interpretability, and enabling the model to learn the markings of potential focal regions during training. This lays the foundation for future fine-grained learning and analysis of focus types. We consider two types of image focus in our approach. The first type is objects within the image, such as people or buildings, which we extract using an object detection model [1], named OD regions. The second type is visual areas in the image that are related to the text, which we locate using a visual grounding model [29], named VG regions. Specifically, we first identify nouns in the text using a noun detection tool [1] and form a sequence of keywords with the given entities in the text. Subsequently, for each keyword in the sequence, we use it as a search guide to obtain the relevant visual areas from the image using the visual grounding model. By augmenting these two types of image focus, we preserve not only the global information in the image but also emphasize focal regions related to the text.

## 3.4 Multimodal Fusion

After obtaining the unimodal representations of text and images separately, multimodal integration is required to combine visual

---
[1] https://github.com/explosion/spaCy

information with the textual features to obtain task-relevant multi-modal representations. The focal attention, gating, and focal fusion modules are key components of the multimodal fusion layer utilized for this purpose.

**Focal Attention Module.** The focal attention module is designed to dynamically adjust the model's attention to different regions of the image by computing the similarity between textual features and visual features. To accomplish this, a cross-modality attention network is utilized to associate words in the text with patches in the image. Specifically, the textual features are treated as queries $Q$ ($h_{text}W_q$), and the visual features are used as keys $K$ ($h_{image}W_k$) and values $V$ ($h_{image}W_v$). The module then calculates the similarity between the queries and each key to obtain weights that reflect the significance of each key. Finally, the attention weights are multiplied with the visual features $V$ to obtain the integrated visual information $\hat{h}_{image}$.

$$\hat{h}_{image} = \text{softmax}\left(\frac{QK^T}{\sqrt{d_k}}\right)V \tag{3}$$

where $d_k$ is the same as the dimension of $h_{text}$ because a single head is used.

This design is advantageous since it enables the model to focus on both local information within the focal regions and global information across the entire image. As compared to directly inputting the entire image or focusing solely on relevant regions, this method provides a more comprehensive approach for capturing associations between text and images, resulting in improved accuracy in relation extraction.

**Gating Module.** In practical applications, the relation extraction task may be negatively impacted by the limitations of visual grounding and object detection technologies, which may result in the integration of irrelevant information that can introduce noise and reduce accuracy. To address this issue, a gating mechanism is proposed to filter out irrelevant image focal regions, enabling the model to adaptively select relevant focal regions that are useful for the task. This effectively reduces the interference of errors from visual grounding and object detection models on the final results. The gating variable $\lambda$ is defined to control how much visual information is retained, with a value range of [0,1]. $\lambda$ is optimized through backpropagation algorithms, making it trainable. Based on the $\lambda$, textual and visual information are fused to generate the final output representation $h_{out}$:

$$\lambda = \text{sigmoid}(U \times h_{text} + V \times \hat{h}_{image})$$
$$h_{out} = (1 - \lambda) \times h_{text} + \lambda \times \hat{h}_{image} \tag{4}$$

where $U$ and $V$ are trainable variables. $\lambda$ controls how much visual information is kept. When $\lambda$ approaches 0, the fused output is closer to the text representation, indicating that the model relies more on textual information. Conversely, when $\lambda$ approaches 1, the fused output is more biased towards image features.

**Focal Fusion.** The focal fusion module integrates text features augmented with various types of focal region information through the focal attention and the gating modules. For each VG region augmented image, we obtain the output representation $h_{out}^{VG}$ from the gating module and integrate all $h_{out}^{VG}$ to form a tensor $H_{out}^{VG}$.

To account for different degrees of importance among the focal regions, we introduce a weight coefficient $w_{vg}$ with a length of $VG_n$. During training, we initialize $w_{vg}$ and dynamically adjust it via backpropagation. This allows us to perform a weighted average of $H_{out}^{VG}$ and obtain the weighted representation $\overline{H}_{out}^{VG}$ as follows:

$$\overline{H}_{out}^{VG} = w_{vg} \times H_{out}^{VG} \tag{5}$$

We apply the same procedure to acquire $\overline{H}_{out}^{OD}$ for images augmented using OD regions.

For relation prediction, the classifier leverages the vectors corresponding to the head and tail entities in $\overline{H}_{out}^{VG}$ and $\overline{H}_{out}^{OD}$. We concatenate these vectors into a single vector, $z$:

$$z = \text{concat}[\overline{H}_{out[head_{start}]}^{VG}, \overline{H}_{out[tail_{start}]}^{VG}, \\ \overline{H}_{out[head_{start}]}^{OD}, \overline{H}_{out[tail_{start}]}^{OD}] \tag{6}$$

here, $head_{start}$ and $tail_{start}$ denote the start position markers for the head and tail entities, respectively.

The vector $z$ is then fed into a multi-layer perceptron (MLP) to predict the relation $r$ :

$$r = \text{argmax}(MLP(z)) \tag{7}$$

## 4 EXPERIMENTS

### 4.1 Dataset

A detailed experimental evaluation was conducted on the MNRE dataset [36], an artificially annotated multimodal relation extraction dataset. The text and images in MNRE were harvested from Twitter and cover multiple thematic domains, including music and sports. This dataset comprises a total of 15,484 samples and 9,201 images, encompassing 23 relation categories. To conduct the experiments, the dataset was divided into training, validation, and test sets, containing 12,247, 1,624, and 1,614 samples, respectively.

**Challenge Set.** To comprehensively evaluate the performance of the model under various types and severities of noise, two challenge sets were constructed: **Align-Noise** and **Aug-Noise**. The Align-Noise challenge set simulates potential misalignment of image-text pairs in practical applications by randomly shuffling them. This tests the model's performance under alignment noisy data. The dataset pairs images with segments of text that do not match from the dataset according to different shuffling ratios, such as 5%, 10%, 20%, and so forth. The Aug-Noise challenge set simulates errors that may occur from visual grounding and object detection models to test the robustness of relation extraction models based on focal augmented images. We randomly selected a specific proportion of all focal regions and for each one, a random operation from the following was performed: 1) Modifying focal regions: the focal regions were replaced with irrelevant image regions that were randomly selected to simulate false positives of visual grounding and object detection models. 2) Deleting focal regions: the focal regions identified by visual grounding and object detection models were ignored to simulate false negatives of the models. We tested various proportions, such as 10%, 30%, 50%, etc.

## 4.2 Baselines

In this study, we compare the performance of multiple types of baseline models on the MNRE dataset.

**Textual relation extraction models. PCNN** [31] combines piecewise maximum pooling and multi-instance learning for text relation extraction. **MTB** [20] uses distant-supervision and places more emphasis on learning relationship features by establishing a pre-training task to compare various relationship vectors.

**Large Language Models. Llama2-13B** [23] is a large-scale language model released by Meta, designed to handle complex natural language processing tasks with high efficiency and understanding. **Baichuan-13B-Chat** [2] is an open-source, commercially available large-scale language model developed by Baichuan Intelligent Technology. **ChatGLM3-6B** [10] is a generation of pre-trained dialogue models jointly released by Zhipu AI and Tsinghua KEG, with many excellent features such as smooth dialogue and low deployment threshold. **Qwen-14B-Chat** [2] is optimized for delivering high-quality, engaging conversations with a deep understanding of language and context, proposed by Alibaba Cloud. **Qwen-VL-Chat** [3] is a specialized variant of the Qwen model that incorporates visual language understanding, enabling it to not only engage in text-based conversations but also comprehend and generate responses based on visual inputs.

**Multimodal relation extraction models. BERT+SG+Att** [36] utilizes the attention mechanism to consider the semantic similarity between the visual graph (i.e., scene graph) and textual contents. **MEGA** [35] designs an efficient graph alignment that considers both structural similarity and semantic consistency between the visual scene graph and text dependency graph structures. **UMT** [30] leverages regional image features to represent objects to exploit fine-grained semantic correspondences based on transformer and visual backbones. **UMGF** [33] represents text and image as a unified multimodal graph, capturing semantic relationships between multimodal semantic units. **ViL-BERT** [19] is a multimodal two-stream model, allowing it to process both visual and textual inputs in separate streams based on the symmetric attention mechanism. **VisualBERT** [15] uses the self-attention mechanism to implicitly align text elements and image areas, showing powerful multimodal semantic understanding capabilities. **HVPNeT** [7] achieves the fusion of hierarchical multi-scale visual features by utilizing image features as pluggable visual prefixes to guide text representation, and implements a dynamic gating aggregation strategy to enhance the model's robustness to irrelevant image noise. **MKGformer** [6] proposes a multi-level fusion method to guide interaction through coarse-grained prefixes and fine-grained correlation-aware fusion modules integrate visual and textual representations.

## 4.3 Experimental Setup

In the experimental settings, we configured the model to use four visual grounding regions and three object detection regions. To ensure consistency, all focal augmented images were uniformly scaled to a size of 224x224 pixels. We utilized the AdamW optimizer [18] with a learning rate of 1e-5 and a weight decay of 0.08 applied to all non-bias parameters to standardize the model and avoid overfitting. BERT-base-uncased from the Hugging Face library was

[2]https://github.com/baichuan-inc/Baichuan-13B

used to process text input, while ViT-B/32 processed visual input. The sentence max length was set to 128, and the number of image patches was set to 49. The focal attention module included one head per module with a 0.1 dropout ratio applied to the attention calculation results to improve model robustness and avoid overfitting. For LLM-related experiments, a random recall approach was utilized as the instance for context learning, and a total of 2 to 5 instances were used. Additionally, we evaluated our model's performance using precision, recall, and F1-score to maintain consistency with existing multimodal relation extraction works.

## 5 RESULT AND ANALYSIS

### 5.1 Main Results

**Table 1: The overall performance of FocalMRE and other state-of-the-art methods on MNRE[3].**

| Model | Precision | Recall | F1-Score |
|---|---|---|---|
| PCNN | 62.85 | 49.69 | 55.49 |
| MTB | 66.00 | 61.56 | 63.70 |
| Llama2-13B | 5.44 | 5.01 | 5.21 |
| baichuan-13B-Chat | 2.93 | 6.79 | 4.09 |
| ChatGLM3-6B | 8.82 | 9.21 | 9.01 |
| Qwen-14B-Chat | 11.68 | 11.15 | 11.40 |
| Qwen-VL-Chat | 11.42 | 13.09 | 12.20 |
| BERT+SG+Att | 62.95 | 62.65 | 62.80 |
| MEGA | 64.51 | 68.44 | 66.41 |
| UMT | 62.93 | 63.88 | 63.46 |
| UMGF | 64.38 | 66.23 | 65.29 |
| ViL-BERT | 65.78 | 61.88 | 63.77 |
| VisualBERT | 64.63 | 61.09 | 62.81 |
| HVPNeT | 86.95 | 83.28 | 85.08 |
| MKGformer | 86.26 | 84.38 | 85.31 |
| **FocalMRE** | **88.85** | **87.19** | **88.01** |

We conduct experiments on the MNRE dataset, and Table 1 illustrates the overall results. The following observations are made: 1) LLM models without fine-tuning are not effective in performing multimodal relation extraction tasks. 2) Multimodal relation extraction models outperform textual relation extraction models. By leveraging both textual and visual information, multimodal models achieve superior extraction performance. This underscores the significance of exploiting multimodal information in relation extraction tasks. 3) Although models such as BERT+SG+Att, MEGA, UMT, UMGF, ViL-BERT, and VisualBERT exhibit adequate performance in multimodal relation extraction tasks, their overall performance is not particularly remarkable. In contrast, models like HVPNeT and MKGformer demonstrate significant performance improvements by introducing visual grounding or object detection techniques to filter image noise. This finding reveals the potential negative impact of image noise on multimodal relation extraction tasks and

---

[3]The experiment is based on all regions provided by the dataset, including up to 4 VG regions and up to 3 OD regions, and the baseline performance results are re-evaluated under this setting.

emphasizes the essential role of image denoising mechanisms in enhancing model performance. 4) The proposed FocalMRE model displays outstanding performance, surpassing all of the baseline models. This result can be attributed to FocalMRE's comprehensive incorporation of local focal information and global image information, coupled with efficient denoising capabilities. Particularly for datasets with high levels of noise, such as social media, the FocalMRE model offers exceptional performance.

To further validate the effectiveness of the FocalMRE model in handling noisy data, we conduct an in-depth comparative analysis with the best-performing baseline model, MKGFormer. The analysis focuses on the challenge set, and the experimental results are presented in Table 2. They reveal the performance of the two models under different noise conditions. Specifically, on the Align-Noise challenge set, both models exhibit decreasing performance as the noise ratio increases. However, the FocalMRE model outperforms MKGFormer, demonstrating its robustness in handling image-text alignment noise. On the Aug-Noise challenge set, the FocalMRE model also displays superior performance over MKGFormer, indicating that even when visual grounding and object detection models produce errors, FocalMRE's denoising mechanism can accurately identify and eliminate these errors, leading to enhanced robustness and stability. These findings provide comprehensive evidence supporting the superiority of the FocalMRE model in multimodal relation extraction tasks, particularly in the presence of noise interference.

**Table 2: Performance comparison between FocalMRE and MKGformer on challenge sets.**

| Dataset | Ratio | MKGformer | | | FocalMRE | | |
|---------|-------|-----|-----|-----|-----|-----|-----|
| | | P | R | F1 | P | R | F1 |
| Align-Noise | 5% | 71.34 | 74.69 | 72.98 | 77.10 | 73.12 | **75.06** |
| | 10% | 74.41 | 69.53 | 71.89 | 76.34 | 71.09 | **73.62** |
| | 20% | 70.97 | 72.19 | 71.57 | 71.80 | 72.81 | **72.30** |
| | 50% | 65.21 | 65.31 | 65.26 | 63.74 | 68.13 | **65.86** |
| | 100% | 62.95 | 62.66 | 62.80 | 66.21 | 60.31 | **63.12** |
| Aug-Noise | 10% | 83.49 | 82.19 | 82.83 | 86.71 | 83.59 | **85.12** |
| | 30% | 79.02 | 75.94 | 77.45 | 83.67 | 78.44 | **80.97** |
| | 50% | 76.90 | 71.25 | 73.97 | 77.42 | 75.00 | **76.19** |
| | 100% | 70.45 | 64.06 | 67.10 | 65.15 | 69.53 | **67.27** |

## 5.2 Ablation Study

To further investigate the precise impact of the focal attention module and gating mechanism on model performance, we conducted an ablation experiment based on the FocalMRE model. The results of this experiment, presented in Table 3, highlight the crucial roles played by the focal attention module and gating mechanism in the multimodal relation extraction task. The experimental data clearly demonstrates that removing either the focal attention module or the gating mechanism results in a significant decrease in model performance. Additionally, when both modules are removed simultaneously, the performance drop becomes even more pronounced. These findings strongly support the synergy between these two

modules in the FocalMRE model: they enhance each other's performance and cooperatively improve the overall performance of the model.

**Table 3: Ablation results.**

| Module | Precision | Recall | F1-Score |
|--------|-----------|--------|----------|
| FocalMRE | 88.85 | 87.19 | 88.01 |
| w/o Focal Attention | 86.03 | 84.96 | 85.35 (-2.66) |
| w/o Gating | 85.80 | 85.94 | 85.87 (-2.14) |
| w/o Either | 84.91 | 82.66 | 83.77 (-4.24) |

## 5.3 Detailed Analysis

This section aims to present a detailed analysis of the performance of the model from multiple perspectives:

*5.3.1 Effect of the focus type and quantity.* In the experiment illustrated in Figure 3, we conduct a detailed analysis of the impact of two types of focal regions on model performance: VG regions detected through visual grounding and OD regions detected through object detection models. The experimental results demonstrate that when the number of OD regions is kept fixed, increasing the number of VG regions can improve the model performance to some extent. More specifically, when the number of VG regions exceeds 2, the performance improvement is even more pronounced. In contrast, when the number of VG regions is fixed, increasing the number of OD regions has a relatively smaller impact on the model performance. These findings indicate that VG regions are more effective in feature extraction and provide richer information to the model. Hence, they are more beneficial in improving model performance.

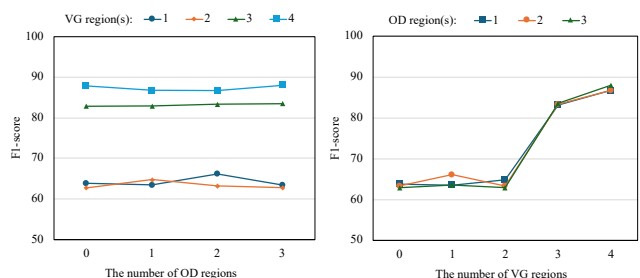

**Figure 3: The impact of the number of different types of focal regions on model performance.**

*5.3.2 Performance comparison on each relation type.* To assess the generalizability of the proposed method across different types and sample sizes of relations, we examined the model's performance on each non-none relation type. Given the potential differences in data distribution, it is crucial to verify whether the model shows consistent performance improvement across different types of relations. As indicated in the Table 4, the model's performance has significantly improved on the majority of relations, particularly for

long-tail relations. The baseline model, on the other hand, offers almost no accurate predictions for such relations, whereas FocalMRE has demonstrated remarkable improvement. Despite these significant improvements, there is still substantial room for enhancing the model's performance on long-tail relations.

**Table 4: Performance comparison on each relation type. F1-score is used as the metric.**

| Relation (size) | MKGformer | FocalMRE |
|---|---|---|
| /per/per/peer (156) | 88.82 | 93.54 (+4.72) |
| /per/org/member_of (110) | 88.70 | 94.12 (+5.42) |
| /loc/loc/contain (99) | 97.98 | 98.00 (+0.02) |
| /per/misc/present_in (74) | 96.05 | 99.33 (+3.28) |
| /org/loc/locate_at (46) | 86.05 | 83.72 (-2.33) |
| /per/loc/place_of_residence (29) | 78.12 | 66.67 (-11.45) |
| /per/per/alternate_names (21) | 54.55 | 72.73 (+18.18) |
| /per/per/couple (19) | 50.00 | 77.42 (+27.42) |
| /misc/loc/held_on (18) | 100.00 | 100.00 (0.00) |
| /org/org/subsidiary (16) | 62.86 | 58.06 (-4.80) |
| /misc/misc/part_of (14) | 0.00 | 25.00 (+25.00) |
| /per/misc/nationality (10) | 95.24 | 95.24 (0.00) |
| /org/org/alternate_names (8) | 0.00 | 31.58 (+31.58) |
| /per/loc/place_of_birth (7) | 0.00 | 15.38 (+15.38) |
| /per/per/parent (4) | 0.00 | 0.00 (0.00) |
| /per/misc/awarded (4) | 0.00 | 85.71 (+85.71) |
| /per/per/neighbor (2) | 0.00 | 0.00 (0.00) |
| /per/per/siblings (1) | 0.00 | 0.00 (0.00) |
| /per/per/charges (1) | 0.00 | 0.00 (0.00) |
| /per/misc/religion (1) | 0.00 | 0.00 (0.00) |

*5.3.3 The impact of global image information.* To investigate the effect of global image information on multimodal relation extraction based on the proposed FocalMRE model, we conducted experiments using three different image input methods: original images, cropped image focal regions, and the proposed image focal augmentation method. The experimental results, as presented in Table 5, demonstrate that when using the original image without pre-processing as the input, the model's performance is significantly lower. However, when the focal region of the image is cropped, the model's performance significantly improves, suggesting that removing irrelevant noise information from the image can effectively enhance the model's performance. Furthermore, using the focal augmentation method, which not only highlights the focal region information but also balances the preservation of global image information and removal of irrelevant noise information through gated mechanisms, leads to further improvements in model performance. This outcome confirms the critical role of global image information in multimodal relation extraction and validates the effectiveness of the image focal augmentation method and gating mechanism in optimizing model inputs.

*5.3.4 The impact of focal augmentation on visual attention.* We conducted a visualization analysis to examine the model's ability to concentrate on the focal areas and their global information during image processing. Figure 4 displays the visualization analysis

**Table 5: Comparison of model performance using different image input methods. The original image contains both global information and substantial noise. Cropping the focal regions from the image eliminates noise and global information. The focal region augmentation approach enhances the focal information while preserving the global information.**

| Input | Precision | Recall | F1-Score |
|---|---|---|---|
| Original image | 64.52 | 61.09 | 62.76 |
| Focal regions cropped | 87.60 | 86.09 | 86.84 |
| Focal augmented | **88.85** | **87.19** | **88.01** |

results for three examples, each comprising three attention visualization images. One image represents the attention visualization on the original image, while the other two depict the attention visualization on the focal augmented images. In particular, one image is designated as the focal augmented image that is strongly correlated with the image-text (upper row), while the other image is weakly correlated (lower row).

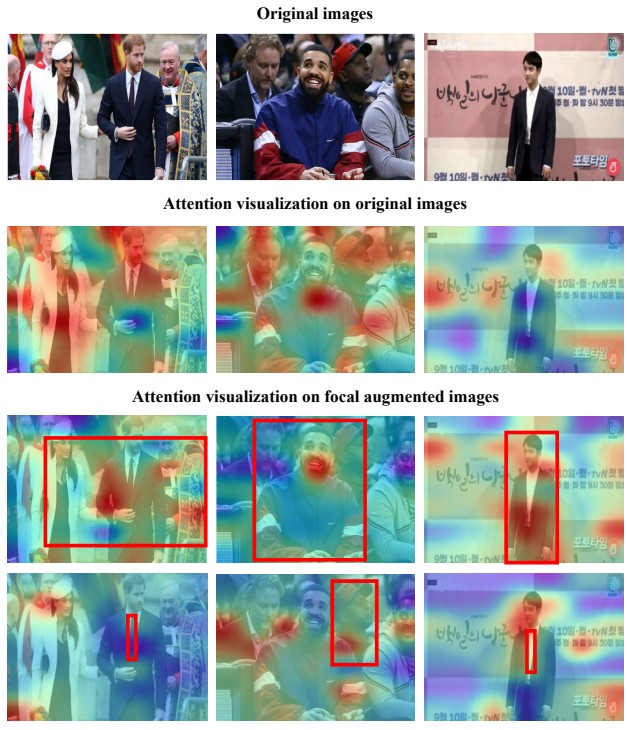

**Figure 4: The attention visualization result. (red indicates high attention weight, blue indicates low attention weight)**

In the example a), the model's attention is distributed across all individuals in the original image. However, when the strongly correlated area with the image-text is augmented (i.e., the central visual area of the image), the model's attention is focused on the vital information within the focal area. This information, such as

the action of holding hands in the image, is beneficial for predicting the relationship type (e.g., /per/per/couple). Remarkably, when the weakly correlated area (i.e., Prince Harry's tie) is augmented, the model does not blindly focus its attention on the tie. Instead, the model's attention is more focused on other people in the surrounding area outside the focus. This contrasting phenomenon is attributed to our approach of combining image focal augmentation with gating mechanisms, which allows the model to selectively focus on the image within the focal area without ignoring the global information in the image. This effectively achieves a balance between important local information and global information in the image. Examples b) and c) also demonstrate similar results, further emphasizing the effectiveness of our method.

*5.3.5 The performance in low-resource scenarios.* To provide a more comprehensive evaluation of the proposed model's performance in low-resource scenarios, we have conducted two types of experiments: 1) Low-resource training set: we sampled the training data in different proportions, namely 5%, 10%, 20%, 30%, 40%, and 50% to investigate the effect of different training data sizes on model performance. 2) Low-resource relation samples: this experiment involved retaining $K$ samples for each relation type to form the training set. By controlling the value of $K$, we aimed to examine how the model's performance varies with limited samples.

**Table 6: Performance comparison in the low-resource training set scenario. The proportions indicate the amount of the training set that is retained.**

| Model | 5% | | | 10% | | |
|---|---|---|---|---|---|---|
| | P | R | F1 | P | R | F1 |
| ViL-BERT | 33.40 | 49.06 | 39.75 | 61.26 | 50.16 | 55.15 |
| HVPNeT | 63.68 | 60.00 | 61.79 | 80.74 | 78.59 | 79.65 |
| MKGformer | 58.80 | 56.73 | 57.75 | 64.54 | 63.12 | 63.82 |
| FocalMRE | **72.50** | **71.25** | **71.87** | **82.91** | **81.09** | **81.99** |

| Model | 20% | | | 30% | | |
|---|---|---|---|---|---|---|
| | P | R | F1 | P | R | F1 |
| ViL-BERT | 61.14 | 55.31 | 58.08 | 65.36 | 54.84 | 59.64 |
| HVPNeT | 84.83 | 81.25 | 83.00 | 84.43 | 82.19 | 83.29 |
| MKGformer | 83.07 | 81.25 | 82.15 | 84.50 | 82.66 | 83.57 |
| FocalMRE | **86.12** | **84.38** | **85.24** | **85.51** | **84.84** | **85.18** |

| Model | 40% | | | 50% | | |
|---|---|---|---|---|---|---|
| | P | R | F1 | P | R | F1 |
| ViL-BERT | 61.34 | 62.97 | 62.14 | 65.77 | 60.94 | 63.26 |
| HVPNeT | 85.83 | 83.28 | 84.54 | 85.76 | 83.75 | 84.74 |
| MKGformer | 84.52 | 83.59 | 84.05 | 85.58 | 84.38 | 84.97 |
| FocalMRE | **87.36** | **86.41** | **86.88** | **87.68** | **86.72** | **87.20** |

From the results in Table 6 and Table 7, we can observe: 1) The impact of noisy information on model performance is more pronounced in low-resource scenarios. The results demonstrate that models designed to handle image noise, such as HVPNeT and MKGformer, showed significant performance improvements in comparison to the pre-trained ViL-BERT model. This indicates that mitigating noise issues is essential for models to tackle various

low-resource scenarios effectively. 2) Moreover, the proposed FocalMRE model consistently outperforms the baseline models across almost all metrics and under various low-resource scenarios. This demonstrates the clear advantages of FocalMRE in handling data scarcity. In contrast to models like MKGformer, FocalMRE mainly aims to achieve a balance between utilizing global information and avoiding noise interference. FocalMRE employs focal augmentation, focal attention, and a gating mechanism to fully utilize both the local focal information and the global information in images without interference from noisy information. This advantage is even more pronounced in low-resource scenarios where information is very limited, posing a significant challenge for models to learn useful information. The effective use of global image information undoubtedly provides the model with a richer set of usable information. It is worth noting that this does not require any external knowledge but rather makes full use of the available data. Experiments have proven that this approach is simple yet highly effective in addressing the problem of information scarcity in low-resource scenarios.

**Table 7: Performance comparison in the low-resource relation samples scenario. $K$ means the number of examples for each relation type in the training set.**

| Model | $K = 1$ | | | $K = 2$ | | |
|---|---|---|---|---|---|---|
| | P | R | F1 | P | R | F1 |
| ViL-BERT | 04.16 | 08.91 | 05.67 | 07.93 | 19.53 | 11.28 |
| HVPNeT | 09.37 | 21.25 | 13.00 | 10.39 | 25.62 | 14.79 |
| MKGformer | 08.80 | 22.19 | 12.60 | 10.35 | 21.25 | 13.92 |
| FocalMRE | **13.60** | **22.66** | **17.00** | **13.01** | **31.09** | **18.34** |

| Model | $K = 5$ | | | $K = 10$ | | |
|---|---|---|---|---|---|---|
| | P | R | F1 | P | R | F1 |
| ViL-BERT | 11.03 | 27.50 | 15.75 | 15.93 | 38.28 | 22.50 |
| HVPNeT | 16.46 | 40.31 | 23.38 | 22.47 | 50.62 | 31.12 |
| MKGformer | 21.29 | **41.25** | 28.09 | 38.18 | **53.75** | 44.65 |
| FocalMRE | **29.54** | 40.94 | **34.32** | **46.57** | 45.62 | **46.09** |

## 6 CONCLUSION

The challenge of noise interference in multimodal relation extraction in social media requires a higher level of accuracy and efficiency. Existing methods that filter image information tend to overlook the crucial global image information. To remedy this issue, we propose a novel focus-and-gating method named FocalMRE, which leverages focal augmentation, focal attention, and a gating mechanism to fully utilize both focal and global information in images without interference from noisy information. Experimental verifications on MNRE, which include various settings such as high noise, long-tail, and low-resource scenarios, demonstrate the superior performance of our method, thereby indicating new and promising ideas for multimodal relation extraction, especially in domains characterized by high levels of noise, such as social media.

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
