# OpenReview forum: "Focus & Gating: A Multimodal Approach for Unveiling Relations in Noisy Social Media"
_acmmm.org/ACMMM/2024/Conference — MM2024 Poster_

### Official Review · Reviewer_HAge · 2024-05-21

**Rating:** 3
**Confidence:** 3

**Summary:**

This paper introduces FocalMRE, a novel approach for multimodal relation extraction in the context of image-text data from social media. FocalMRE utilizes image focal augmentation, focal attention, and gating mechanisms to enhance model performance. By focusing on both focal regions and global image information, the model effectively addresses noise interference. Experimental results on the MNRE dataset demonstrate the method's robustness and significant performance advantages, particularly in scenarios with high noise levels, long-tail distributions, and limited resources.

**Strengths:**

1. FocalMRE employs image focal augmentation and focal attention mechanisms to concentrate on relevant regions within images, enhancing the effectiveness of multimodal relation extraction.

2. Gating mechanisms for noise reduction: FocalMRE incorporates gating mechanisms to optimize multimodal fusion, enabling the model to selectively emphasize the most relevant augmented regions.

3. The ablation study seems sufficient and shows the effectiveness of the designed modules.
4. Clear and well-motivated reasoning in the paper. Well-written and structured.

**Limitations:**

1.  It's best to be able to provide the demo and code for checking. I would be glad to change my score if the code and data will be open sourced.
2. Only one public dataset is used to validate.
3. Can you elaborate on the technical workings of the Gating Module and how it helps?
4. The object detection model and the visual grounding model used are too old.
5. Don't quite understand where global image features are used, only local image features are seen in Figure 2. And the Focal Fusion module is not clearly indicated in the Figure 2 leading to confusion

**Suitability:**

3

---

### Official Review · Reviewer_zhF7 · 2024-05-21

**Rating:** 3
**Confidence:** 3

**Summary:**

This paper proposes a multimodal relation extraction method called FocalMRE. FocalMRE utilizes image focal enhancement, focal attention and gating mechanisms to focus on the focal region of an image while integrating global image information. Extensive experiments validate the effectiveness of the method.

**Strengths:**

1.The paper is well-organized and clearly written.

2.The FocalMRE employs focal enhancement and gating mechanisms to optimize the multimodal fusion strategy, thereby mitigating noise interference and improving the accuracy of relation extraction.

3.Extensive experiments on the MNRE dataset prove the effectiveness of the proposed method, while the FocalMRE model still maintains a good detection performance in low-resource scenarios.

**Limitations:**

1.For Table 1, I reviewed some recently published papers on multimodal relation extraction, such as the references below, and the following questions arose:

(1) Why are the baselines HVPNeT and MKGformer performances in Table 1 improved so much over the same baseline performances given in the reference papers I provided? They even outperform all the proposed methods in the references.

(2) The baselines compared in multimodal relation extraction models are somewhat outdated, what would be the performance if one or more of the reference papers are used as new baselines?

2.Since FocalMRE needs to use multiple external models for image enhancement, and at the same time the enhanced images all need to go through the visual encoder, will this have a greater impact on the computational efficiency of FocalMRE.

3.If you can answer my doubt, I'll consider raising my score.

References:

[1] Hu X, Chen J, Liu A, et al. Prompt me up: Unleashing the power of alignments for multimodal entity and relation extraction[C]//Proceedings of the 31st ACM International Conference on Multimedia. 2023: 5185-5194.

[2] Hu X, Guo Z, Teng Z, et al. Multimodal Relation Extraction with Cross-Modal Retrieval and Synthesis[C]//Proceedings of the 61st Association for Computational Linguistics (Volume 2: Short Papers), 2023: 303–311.

[3] Liu X, Hu C, Zhang R, et al. Multimodal Relation Extraction via a Mixture of Hierarchical Visual Context Learners[C]//Proceedings of the ACM on Web Conference 2024. 2024: 4283-4294.

**Suitability:**

3

---

### Official Review · Reviewer_uyPP · 2024-05-23

**Rating:** 3
**Confidence:** 4

**Summary:**

This paper proposes FocalMRE to address the noisy information problem in MRE. The method achieves SOTA performance compared to exsiting methods.

**Strengths:**

1.Good number of experiments and ablation studies have been shown.
2.The results show considerable performance over the peer works.

**Limitations:**

1.Lack of novelty, the focal attention is confusing. It seems focal has the same meaning with attention.
2.The writing of this paper is poor.
3.It would be better to test the model on more tasks such as MNRE.

**Suitability:**

3

---

### Official Review · Reviewer_c1Pb · 2024-05-24

**Rating:** 5
**Confidence:** 3

**Summary:**

This paper proposes a novel multimodal relation extraction method, named FocalMRE, which leverages image focal augmentation, focal attention, and gating mechanisms. The method effectively utilizes the global information in the image and can select the most relevant augmented regions for overcoming noise interference in relation extraction.

**Strengths:**

- The paper is well written and easy to understand.
- The performance of the proposed method is promising compared with state-of-the-art approaches. Ablation study and visualization are sufficient to reason about the proposed approach.

**Limitations:**

The proposed method requires multiple encoder forward passes to classify a sample. How is it compared to other baselines regarding computational efficiency?

**Suitability:**

3

---

### Meta-Review · Area_Chair_gDAd · 2024-07-04

**Recommendation:** Accept (Poster)
**Confidence:** 3

**Metareview:**

The reviewers generally agree that the paper is well-structured and addresses an important problem in multimodal relation extraction. They highlight the method's innovative use of image focal augmentation, focal attention, and gating mechanisms to enhance performance in relation extraction tasks. The extensive experimentation, particularly the ablation studies and visualizations, was noted as a strength by multiple reviewers, demonstrating the effectiveness of the proposed approach. Therefore, my recommendation for this paper is Accept. I encourage the authors to integrate the reviewer's feedback into the final version.